# Roles of Sodium Hydrogen Exchanger (NHE1) and Anion Exchanger (AE2) across Chondrocytes Plasma Membrane during Longitudinal Bone Growth

**DOI:** 10.3390/membranes12070707

**Published:** 2022-07-14

**Authors:** Adamu Abdul Abubakar, Ahmed Khalaf Ali, Sahar Mohammed Ibrahim, Kareem Obayes Handool, Mohammad Shuaib Khan, Noordin Mohamed Mustapha, Tengku Azmi Tengku Ibrahim, Ubedullah Kaka, Loqman Mohamad Yusof

**Affiliations:** 1Department of Companion Animal Medicine and Surgery, Universiti Putra Malaysia, Serdang 43400, Malaysia; abubakar.adamu@udusok.edu.ng (A.A.A.); dr.ahmadkhalaf78@yahoo.com (A.K.A.); sahar2011973@yahoo.com (S.M.I.); handoolk_k59_o@yahoo.com (K.O.H.); shoaibbaloch2002@yahoo.com (M.S.K.); dr_ubedkaka@upm.edu.my (U.K.); 2Department of Veterinary Surgery and Radiology, Usmanu Danfodiyo University, Sokoto PMB 2346, Nigeria; 3Department of Surgery and Theriogenology, College of Veterinary Medicine, University of Mosul, Mosul 00964, Iraq; 4Faculty of Veterinary and Animal Science, Gomal University, Dera Ismail Khan 29050, Pakistan; 5Department of Veterinary Pathology and Microbiology, Universiti Putra Malaysia, Serdang 43400, Malaysia; noordinmm@upm.edu.my; 6Department of Pre-Clinical Veterinary Sciences, Universiti Putra Malaysia, Serdang 43400, Malaysia; tengkuazmi@upm.edu.my

**Keywords:** long bone growth, growth plate, chondrocytes, plasma membrane inhibitors

## Abstract

Mammalian long bone growth occurs through endochondral ossification, majorly regulated by the controlled enlargement of chondrocytes at the growth plate (GP). This study aimed to investigate the roles of Na^+^/H^+^ (sodium hydrogen exchanger (NHE1)) and HCO_3_^−^ (anion exchanger [AE2]) during longitudinal bone growth in mammals. Bones from P10 SpragueDawley rat pups were cultured exvivo in the presence or absence of NHE1 and AE2 inhibitors to determine their effect on long bone growth. Gross morphometry, histomorphometry, and immunohistochemistry were used to assess the bone growth. The results revealed that the culture of the bones in the presence of NHE1 and AE2 inhibitors reduces bone growth significantly (*p* < 0.05) by approximately 11%. The inhibitor significantly (*p* < 0.05) reduces bone growth velocity and the length of the hypertrophic chondrocyte zone without any effect on the total GP length. The total GP chondrocyte density was significantly (*p* < 0.05) reduced, but hypertrophic chondrocyte densities remained constant. NHE1 fluorescence signaling across the GP length was higher than AE2, and their localization was significantly (*p* < 0.05) inhibited at the hypertrophic chondrocytes zone. The GP lengthening was majorly driven by an increase in the overall GP chondrocyte and hypertrophic chondrocyte densities apart from the regulatory volume phenomenon. This may suggest that NHE1 and AE2 could have a regulatory role in long bone growth.

## 1. Introduction

Mammalian bone growth in length follows the process of endochondral bone formation through sequential chains of chondrocyte proliferation and differentiation at the growth plate (GP) [1]. The endochondral ossification (EO) process is highly regulated to ensure both normal neonatal bone development and postnatal long bone growth at the epiphyseal growth plate (EGP) [2,3]. During fetal bone development, mesenchymal germinal cells migrate to form stem-like chondroprogenitor cells, which later undergo differentiation into chondrocytes and extracellular matrix (ECM) with abundant collagen type II and aggrecan [4,5]. At the stage of postnatal bone elongation and development, the pre-formed chondrocytes in the epiphyseal growth plate (GP) undergo an ordered andhighly regulated cascade of the cell cycle, involving proliferation, maturation, hypertrophy, and death of chondrocytes at the chondro-osseous junction [4,6]. This differentiation cascade of events is regulated by paracrine and other signaling pathways (Wnt signaling, Indian hedgehog (IHH), transforming growth factor (TGF-β), bone morphogenic protein (BMP), vesicular endothelial growth factor (VEGF), fibroblast growth factor (FGF), and platelet-derived growth factor (PDGF)), which bring about linear bone growth in mammals [7,8].

The mammalian growth plate is an organized thin layer of cartilage entrapped between the epiphysis and metaphysis at the end of the long bone [9,10]. Histologically, the GP is composed of five principal layers: the resting, proliferative, early and late hypertrophic, and early mineralized layers [11,12]. The resting zone contained stem-like chondroprogenitor cells that were believed to have a limited exhausted differentiation and proliferative capacity in all mammals except rodents. The finite proliferative capacity brings about growth plate fusion and the subsequent cessation of long bone growth, the cells in this zone are scattered within the cartilage matrix and usually have low rates of proliferation [5,6]. Proliferative zone chondrocytes are very active in terms of mitotic division, and they play a significant role in endochondral ossification. The chondrocytes in this zone are arranged in columns along the long axis of the bone [3,13]. The early and late hypertrophic chondrocytes zones (EHCZ and LHCZ) contain matured chondrocytes, following terminal differentiation of the proliferative chondrocytes [14]. Morphologically, the chondrocytes located in these two zones are distinguished based on the dimension of their height. The early and late hypertrophic chondrocytes were reported to be ≤7µm and ≥9 µm in height, respectively [15,16]. Hypertrophic chondrocytes play the most significant role in the endochondral ossification, at a certain period of their proliferative cycle; these cells cease mitotic division, enlarge, and subsequently die and become invaded by the extracellular matrix and adjacent blood vessels. The invaded structures then differentiate into osteogenic cells, hence the formation of new bone tissue that brings about an increase in bone length [5,11].

The mechanism by which hypertrophic chondrocytes die is still debatable, but it was predominantly reported that they die through programmed cell death (apoptosis). However, some studies reported autophagy and cell hypoxia as the mechanism of the cell death [17,18]. Similarly, the mechanism through which the proliferative chondrocytes differentiate and mature into hypertrophic chondrocytes is poorly understood. However, Loqman et al. [10] and Bush et al. [19] had earlier reported that plasma membrane transporters are implicated through the osmotic regulation of chondrocytes volume hypertrophy due to ion exchanges within the hypertrophic chondrocyte boundaries. Moreover, Cooper et al. [2] have reported three different possible ways in which mammalian hypertrophic chondrocytes undergo volume increase that subsequently contribute to long bone growth: (1) by a proportionate increase in dry mass production and fluid intake known as true hypertrophy, (2) by chondrocytes swelling, and (3) through a proportionate increase in dry mass along with fluid volume increase.

It is equally important to highlight the regulatory contribution of these transporters that have been already established on other cell types other than chondrocytes during the physiological or pathological process as reported by Malo and Fliegel, [20]; Coppini et al. [21]; and Alka and Casey, [22]. Increasing the localization of NHE1 and AE2 either individually or in combination may promote hypertrophy for example in kidney cells, as reported by Zhao et al. [23], spermiogenesis regulation by Medina et al. [24], hepatocellular carcinoma cells by Liu et al. [25] and Hwang et al. [26], ventricular myocyte by Fliegel, [27], pulmonary artery smooth muscle cells by Yu and Hale, [28], gastric cancer cells by Hosogi et al. [29], the regulation of airway epithelial cells by Kim et al. [30], mammary branching morphogenesis by Jenkins et al. [31], the regulation of biliary epithelial cells by Concepcion et al. [32], and renal tubule epithelial cells by Valles et al. [33].

The main objective of this study is to determine the possible roles of the NHE1 (antiporter) and AE2 (anion exchanger) plasma membrane proteins in longitudinal bone growth, in addition to their regulatory hypertrophic chondrocytes volume phenomenon that has been documented. This is because previous studies conducted by Loqman et al. [10] and Bush et al. [19] had reported that NKCC1, HNE1, and AE2 membrane transporters are involved in longitudinal bone growth ex vivo, through hypertrophic chondrocytes volume regulation. However, it is not known whether an increase in chondrocyte density across the GP has any influence on bone growth. Therefore, this study will focus on the contribution of chondrocyte densities/population within the GP, apart from the changes in chondrocytes volume. To achieve this objective, the ex vivo model of the longitudinal metatarsal and tibial bone growth from P10 rat pups was used. The age of the P10 rat pups was chosen based on our previous study [34]. The bones were cultured for the period of 48 h ex vivo in the presence or absence of the specific pharmacological plasma membrane inhibitors for NHE1 and AE2. The plasma membrane inhibitors used were; 5-(N-ethyl-N-isopropyl) amiloride (EIPA) and 4,4-diiodothiocyano-2,2-stilbenedisulphonate (DIDS) as inhibitors of NHE1 and AE2, respectively. We hypothesized that Na^+^/H^+^ and HCO_3_^−^ plasma membrane transports of hypertrophic chondrocytes have a role in mediating longitudinal bone growth through other mechanisms of cellular distribution apart from the hypertrophic chondrocyte’s cell volume regulation.

## 2. Materials and Methods

### 2.1. Biochemical Solutions Used

Bone dissection media consisting of phosphate buffer saline (PBS from Fisher Bioreagent, Pittsburgh, PA, USA), 7.5% *v*/*v* α-modified essential medium (α-MEM from NaclaiFesque Inc., Tokyo, Japan), 2% *w*/*v* bovine serum albumin (BSA from NaclaiFesque Inc., Tokyo, Japan) prepared at 7.4 pH. The standard culture medium used was α-MEM supplemented with sodium glycerol biphosphate powder (Sigma-Aldrich, Saint Louis, MO, USA) (1 mM), L-ascorbic acid powder (Fisher Scientific, Loughborough, England, UK) at 5 mgmL^−1^ and penicillin–streptomycin antibiotic (Sigma-Aldrich, USA) at a final concentration of 100 IUmL^−1^ and 100 µg mL^−1^, respectively. Bone tissues were fixed in a combination of 1.3% glutaraldehyde (GA) and 0.5% ruthenium hexamine trichloride (RHT), and the histological slides were stained with 0.1% toluidine blue O, in PBS at pH 5.6 (Acros organics, Morris Plains, NJ, USA). The plasma membrane inhibitors used were: (5-(N-ethyl-N-isopropyl) amiloride [(EIPA) and (4,4-diiodothiocyano-2,2-stilbenedisulphonate) (DIDS), all from Sigma-Aldrich, USA. EIPA and DIDS were freshly prepared for each experiment as 120 mM and 100 mM stock solutions in Dimethyl sulphate (DMSO) (Fisher Scientific, UK) and 0.1 M Potassium bicarbonate (KHCO_3_) (Acros organics, Morris Plains, NJ, USA) respectively. Primary antibodies used were: rabbit polyclonal IgG to NHE1 (sc-28758) and AE2 (sc-99048) from Santa Cruz Biotechnology, Inc., Dallas, TX, USA, for detection of tissue localization of Na^+^/H^+^ and HCO_3_^−^ respectively, while the secondary antibodies used were; goat anti-rabbit IgG pre-packed in IMMUNO CRUZ rabbit ABC staining system kit for immunoperoxidase staining (IP) and goat anti-rabbit IgG-CFL conjugated with fluorescence (Santa Cruz Biotechnolgy, Inc., Dallas, TX, USA). A solution of 10 mM sodium citrate buffer at pH 6.0 (Sigma-Aldrich, USA) and Tween 20 (DAKO North America Inc., Carpinteria, CA, USA) was used for antigen retrieval. H_2_O_2_ 0.3% (Sigma-Aldrich, USA) was used to quench endogenous tissue peroxidase. A Liquid blocker super PAP pen (Daido Sanyo Co., Ltd., Tokyo, Japan) was used to demarcate tissue sections on the slides. Pentobarbital injection at 200 mg/mL (Dolethal; Vetoquinol, France) was used for rat euthanasia.

### 2.2. Animal Preparationand Grouping

Twenty (20) female Sprague Dawley rat pups 10 daysold (P10) were used in the study. The choice of the P10 pups was based on our previous study [20]. The pups were randomly divided into four treatment groups with (*n* = 5) each; EIPA and DIDS plasma membrane inhibitors treatments at respective concentrations of 444 µM and 250 µM, respectively, and their respective controls (DMSO solution as EIPA vehicle and KHCO_3_ solution as DIDS vehicle). The choice of these concentrations was based on the previous study conducted by Loqman et al. [10] where optimal inhibitory concentrations were determined. The rats were humanely sacrificed using 20% pentobarbital (Dolethal; Vetoquinol, France) at 90 mgkg^−1^ intraperitoneally. The left and right tibia and corresponding three middle metatarsals of each limb were carefully dissected along with intact proximal and distal articular cartilages with the help of a dissecting stereomicroscope (Huvitz; HSZ-645TR, Gyeonggi-do, South Korea). The dissected limbs were temporarily placed in dissecting medium containing (PBS, α-MEM; 7.5% *v*/*v* and BSA; 1 mM) at 7.4 pH to maintain bone viability before incubation as described by Abubakar et al. [34].

### 2.3. Tibia and Metatarsal Growth Length and Velocity Measurements

Digital images of the bones were captured and measured at the baseline (0 h), and 48 h after ex vivoincubation (Figure 1). The length of the harvested tibia and metatarsal rudiments were measured with the VIS plus ver.3.50 image analysis and measurement software (Biovis, ON, Canada) as described by Abubakar et al. [34]. The dissecting stereomicroscope was fitted to a 3.1 megapixels digital camera (VIS imaging; UC3010; Malaysia) connected to a personal computer (PC) via USB cable. The lengths of the bones were measured in centimeters (cm) at X6.5 objective magnification. Images were analyzed as previously described by Martensson et al. [35]. The overall bone length was measured before and after incubation at 0 h, and 48 h. The bone lengths measured were expressed as percentage changes from the baseline length obtained to determine the percentage bone growth rate.

The left and right tibial bones were placed individually in a 6-well tissue culture plate (flat bottom cell culture plate with lid; Sigma-Aldrich, USA), while the metatarsal bone rudiments were cultured individually in a 24-well cell culture plate (NUCLON, delta surface, China). The bones were incubated for a period of 48 h in a standard bone culture medium at (5% CO_2_, 95% air, pH 7.37, and 37 °C). The media was discarded and changed after every 24 h period of ex vivo incubation. After 48 h time point, bone growth velocity was calculated by dividing the increase in bone length since the previous measurement by the time interval between the two measurements, the values of the growth velocity were expressed in (µm day^−1^) as described by Chargin et al. [16].

### 2.4. Preparation of Growth Plate (GP) for Histology

The bones were fixed overnight in a combination of 1.3% glutaraldehyde (GA) and 0.5% freshly prepared ruthenium hexamine trichloride (RHT) at pH 7.4 to avoid fixative-induced artifacts to chondrocytes’ shape and volume as described by Loqman et al. [36]. The bone tissues were then dehydrated through a series of ethanol solutions using (TP 1020 semi-closed benchtop tissue processor (Leica, Singapore). The tissues were then embedded in paraffin wax using the standard procedure described by Dettmeyer, [37]. The specimens were then cut into longitudinal 5 µm serial sections using Reichert-Jung 2045 (MulticutRotary Microtome, Leica Biosystems Nussloch GmbH, Nussloch, Germany). After de-paraffinization with xylene and rehydration with decreasing concentrations of ethanol solution (100% and 70%), the sections were stained with 0.1% toluidine blue O in PBS (pH 5.6, 30 s, 23 °C) using a technique adopted from Pastoureau et al. [38]. The sections were then mounted on poly-L-lysine coated microscope slides (Menzel-Glaser, Braunschweig, Germany) rinsed briefly in distilled water, and air-dried before mounting with a coverslip.

### 2.5. Quantitative Histology

The histological images of the proximal and distal GPs were taken using an inverted fluorescence microscope (Nikon Eclipse Ti-S, Tokyo, Japan) fitted with a 20× (numeric aperture = 0.5 WD (82,000 µm)) dry objective lens. In order to determine the beginning and terminal end of the GPs, several subjective criteria were used based on cell sizes and organization as described previously by Wilsman et al. [39]. The GP length and different histological chondrocytes zones were identified and measured using an established procedure described by Bush et al. [19]. Briefly, five zones of the GP were identified physically and marked with a drawn freehand line: the resting zone at the top of proliferating cell border, the proliferating chondrocytes zone, early hypertrophic chondrocytes, late hypertrophic chondrocytes and the zone of mineralization at the base of the hypertrophic zone. The shape and size of the early hypertrophic chondrocytes were smaller compared to those of the late hypertrophic chondrocytes; the early hypertrophic chondrocytes were fully nucleated while the majority or all the late hypertrophic chondrocytes were not nucleated because of possible apoptosis induced changes [17,40]. Hypertrophic chondrocytes were defined by a height range of 7–13.48 µm. The late hypertrophic chondrocytes were further defined as the cells in the last lacuna that were not invaded by metaphyseal blood vessels [15,16]. The total length of the GP and the total GP chondrocytes density at proliferative, early, and late hypertrophic zones were measured and counted. The measurement and counting of the chondrocytes were conducted using NIS-Element BR4.20.00 64-bit analytical software (Nikon, Tokyo, Japan) fitted with Nikon digital sight camera DS-Fi2, K16850 (Nikon Corporation, Japan). The GP images were aligned so that the direction of growth was vertical on the computer screen before image capture. The height of the GP and each growth zones were obtained by delineating the top of the growth plate, the junctions between resting and proliferative zones and between proliferative and hypertrophic zones based on the morphological characteristic of the chondrocytes, as well as the chondrosseous junction based on changes in matrix staining. The vertical height of the total GP was measured at five equidistance locations per section. The cell density was determined based on the total cell numbers counted over atleast three measured areas of interest (cells/mm^2^) per slide as previously described by Loqman et al. [10].

### 2.6. Tissue Immunohistochemistry

The immunoperoxidase (IP) staining technique was performed according to IMMUNO CRUZ kit (Santa Cruz Biotechnology, Inc., USA) and recommended protocol adopted from Renshaw [41]. Briefly, after tissue deparaffinization, rehydration, and several washings in PBS, the tissue sections were demarcated with a liquid blocker super PAP pen. Antigen retrieval was performed in 10 mM sodium citrate buffer pH 6.0 at 95 °C for 20 min using a laboratory microwave. The slides were immersed in PBS, containing 0.3% H_2_O_2_, for 30 min to quench tissue endogenous peroxidase. The sections were then incubated with rabbit polyclonal primary antibodies NHE1 and AE2 against Na^+^/H^+^ and HCO_3_^−^ (dilution 1:400) for both antibodies overnight at room temperature in a humidified chamber. PBS was used as the negative control antibody on some selected tissue sections. After several wash steps in PBS, the sections were incubated with a biotinylated goat anti-rabbit secondary antibody. Antibody bindings were visualized using 3,3′-diaminobenzidine (DAB) substrate buffer incubated for 10 min at 37 °C. The sections were counterstained with hematoxylin for 1 min, and then finally rehydrated in increasing concentrations of alcohol.

For the fluorescence immunohistochemistry (FIHC), the tissue sections were exposed to ultraviolet (UV) light for 45 min to bleach tissue auto-fluorescence before deparaffinization. Antigen retrieval and primary antibody incubations were the same as described in the IP protocol above. The sections were incubated with the goat anti-rabbit IgG conjugated fluorescence secondary antibody (dilution 1:200) in a dark humidified chamber for 30 min at 37 °C. The slides were washed several times with PBS in the dark, mounted with Dako anti-fade fluorescent mounting medium, and the edges of the slides were sealed with a clean nail polish as described by Renshaw [41]. The immunoperoxidase and fluorescence labeling was scored by two independent assessors blinded to the experiment design using 5 points scoring criteria described by Fechenko and Reiferath [42]: negative reaction (0); less than 5% positive stain (1); 5–50% positive stain (2); greater than 50% but weak stain (3); and greater than 50% strong stain (4). At least four fields of view were randomly selected from each slide and the localization of membrane proteins was observed and scored within the surrounding chondrocytes.

### 2.7. Data Analysis

Data were presented in the form of tables and graphs. Comparison between two sets of data was carried out using Student’s *t*-test or a suitable non-parametric test if the data set were not normally distributed. Time-course experiments were analyzed with a repeated measure two-way ANOVA for which appropriate post-hoctests for multiple comparisons were conducted. The analysis was carried out using IBM SPSS for Windows, version 22.0, while graphs were plotted using GraphPad Prism 7. Data were presented as the mean and standard error of the mean (SEM). *p* values of <0.05 were considered significant.

## 3. Results

### 3.1. Effects of the Membrane Inhibitors on the Whole Bone Length and Growth Rate and Velocity

The metatarsal and tibial lengths were measured at the baseline and after 48 h incubation ex vivo (Table 1; Figure 1). There were significant metatarsal and tibial length differences between the baseline bone length at 0 h incubation and their corresponding bone length at 48 h after incubation in all the groups except in the EIPA-inhibited treated group of both metatarsal and tibia (Figure 2). The percentage rate of bone growth in both control vehicles (DMSO and KHCO_3_) was higher than those of the respective EIPA-and DIDS-inhibited treated groups.

The metatarsal length growth pattern revealed significant differences between EIPA-treated medium and its control and DIDS-treated medium and its control (unpaired *t*-test; *p* < 0.05). However, there was no significant bone growth difference between EIPA and DIDS even though the growth rate of EIPA-treated bones was more suppressed by the effect of the growth inhibitor (EIPA; 0.89 ± 0.00 cm versus DIDs; 0.92 ± 0.02 cm) and (EIPA; 2.08 ± 0.05 cm versus DIDS; 2.14 ± 0.02 cm) metatarsal and tibia, respectively. A similar pattern of bone growth was also observed in the tibia between the bone growth lengthening in the treatment medium groups and those in the control medium groups.

The metatarsal and tibial length growths (cm) were expressed as percentage growth rate changes (Table 1).

The mean metatarsal percentage growth rate in the EIPA and DIDS controls medium was 18.12 ± 1.93% and 16.34 ± 1.49%, respectively (Table 1). Exposure to EIPA at a 444 µM concentration inhibits the metatarsal percentage growth rate significantly to 5.50 ± 0.96%, while exposure to DIDS at a 250 µM concentration reduces the metatarsal percentage growth rate significantly to 7.43 ± 1.41%. A one-way ANOVA showed significant differences between EIPA-treated and its corresponding DMSO control; there were also significant differences between DIDS-treated and its KHCO_3_ control. There was neither a significant difference (*p* > 0.05) between the DMSO and KHCO_3_ controls nor between EIPA and DIDS-treated groups.

The mean tibial percentage growth rates in the DMSO and KHCO_3_ control medium were 15.63 ± 5.54% and 16.13 ± 3.485%, respectively. Exposure to EIPA at a 444 µM concentration inhibitsthe tibial percentage growth rate to 4.56 ± 0.75%, while exposure to DIDS at a 250 µM concentration reduced the tibial percentage growth rate to 7.54 ± 1.15%. There was a significant percentage of tibial bone growth differences between EIPA-treated and its DMSO control and likewise between the DIDS-treated and its KHCO_3_ control. No significant difference was observed between the EIPA control and the DIDS control.

The metatarsal growth velocity showed remarkable growth rate inhibitions per day in both EIPA (5.14 ± 0.19 µm day^−1^) and DIDS (7.36 ± 1.54 µm day^−1^) when compared to their respective control groups; DMSO control (19.86 ± 2.39 µm day^−1^) and KHCO_3_ control (17.71 ± 1.84 µm day^−1^). There was a significant growth velocity difference between EIPA and its DMSO control and also between DIDS and its KHCO_3_ control (Figure 2). However, there were no significant differences between the DMSO control versus the EIPA-treated and equally between the EIPA-treated versus the DIDS treated.

A remarkable growth velocity per day was also recorded in tibial bone growth, with the EIPA treatment having the highest growth velocity inhibition rate (9.90 ± 1.62 µm day^−1^) when compared to the DIDS-treated group (16.88 ± 2.70 µm day^−1^). There was no significant difference in the tibial growth velocity between the EIPA-treated and its corresponding control, but a significant difference in the DIDS-treated and its control was observed (Figure 3). As observed in metatarsal, there were also no significant differences in tibial growth velocity between the EIPA control versus the DIDS control and the EIPA-treated versus the DIDS-treated groups (Figure 3).

### 3.2. Growth Inhibitory Effects on GP Length, HCZ Length, and Chondrocyte Density

The histomorphometric changes were examined after 48 h ex vivo incubation of the tibial bones following treatments with EIPA and DIDS plasma membrane inhibitors. Emphasis was made on the GP hypertrophic zone (HCZ) chondrocytes. The total length of the GP (µm) and the HCZ length (µm) were measured. The percentage of the HCZ length was determined from the total GP length (Table 2).

There were no significant differences in the total GP length between all the treated groups and their respective controls. Although tibia exposure to EIPA and DIDS at a respective concentration of 444 µM and 250 µM have inhibited the height of the GP length (EIPA; 516 ± 30 µm); (DIDS; 510 ± 29 µm) when compared to their respective control groups (DMSO; 632 ± 56 µm); (KHCO_3_; 603 ± 33 µm). The length of the hypertrophic GP in both EIPA and DIDS was significantly different from their respective control tibial bone. However, when the HCZ length was expressed as a percentage change from the total GP length, there was no significant difference in the percentage HCZ length among all the treated groups.

The tibia subjected to different treatments showed a significant difference in the total GP chondrocyte densities among the treatment groups (Table 3). The hypertrophic chondrocyte density of the GP also shows significant differences among the treatment groups (Table 3; Figure 4).

### 3.3. Immunoperoxidase and Immunofluorescence Labeling of NHE1 and AE2 in Tibial GP

The immunoperoxidase and immunofluorescence labeling of NHE1 and AE2 were examined and scored after 48 h ex vivo incubation of tibial bones. Proximal GP sections were incubated with NHE1 and AE2 primary antibodies to determine the localization of N^+^/H^+^ and HCO_3_^−^ in the GP chondrocytes as described in the Materials and Methods. The immunoperoxidase and immunofluorescence reactive labeling was compared and scored in all the treated groups (Figure 5a,b).

The tibial bone subjected to EIPA at a 250 µM concentration showed a remarkable reduction in labeling intensity scores of both NHE1 and AE2 antibodies when compared with their DMSO or KHCO_3_ control vehicles (1.00 ± 0.12 and 3.5 ± 0.20, respectively) (Table 4). There was a significant difference among the median scores of different treatment groups (nonparametric Kruskal–Wallis one-way ANOVA *p* < 0.05). A significant reduction in the median labeling intensity score of NHE1 and AE2 was also recorded in the DIDS treatment group when compared to their respective control groups. The EIPA-treated group appeared to have lower median labeling scores of both antibodies when compared with the DIDS-treated group.

## 4. Discussions

The longitudinal growth rate of P10 metatarsal and tibial bones of the experimental rats was remarkably inhibited following treatment with EIPA and DIDS plasma membrane inhibitors of NHE1 and AE2. The two inhibitors (EIPA and DIDS) are widely used against membrane transport of NHE1 and AE2, respectively, in many cell types such as neurons, nephrons, hepatic, and cardiac [20,21]. The mechanism of the targeted ions transport inhibitions of both EIPA and DIDS appear to be similar, or they probably share certain similar features, because both pharmacological agents significantly suppressed the growth of bone length under investigation in a relatively similar pattern. However, EIPA appeared to have more suppressive growth effects on the entire bone length, GP hypertrophic zone length, and total GP chondrocyte density when compared to the DIDS effect.

EIPA was identified as an NHE1 inhibitor, with inhibitory effects on different ion channel exchangers, it acts by catalyzing an electroneutral exchange of extracellular sodium and intracellular hydrogen, thereby suppressing the cellular exchange of Na^+^ [27,33,40]. Physiologically, NHE1 regulates pHί, pHe, and cell volume, leading to cell proliferation, growth, migration, and possible apoptosis. However, some researchers reported NHE1 involvement in some pathological processes, particularly in cancer cells and heart failure [29,43].

DIDS is also a well-known anion exchanger (AE2) inhibitor, previously classified as potent chloride channel blockers, but not specific to AE2 inhibitor, but recently it has been re-classified as a potent specific inhibitor of both chloride and bicarbonate ions inhibitors [44,45]. It acts by blocking anions channels reversibly. Recently, it was established to have a possible inhibitory effect on caspase as an apoptosis marker at higher concentrations (500 µM). AE2 is involved in the intracellular regulation of pHί, cell volume, apoptosis, and membrane potential by catalyzing the reversible exchange of (Cl^−^) (HCO_3_^−^) across the plasma membrane in response to cellular environmental changes [25].

The inhibitory effect of EIPA on the whole metatarsal and tibial bone length appeared to be more pronounced compared to that of DIDS (Table 1 and Table 2), which might suggest the significant effect of NHE1 exchange across the chondrocytes’ plasma membrane. The growth rate suppression due to the effect of EIPA on the whole bone clearly shows apparent variation between the baseline parameters as compared to after 48 h incubation. DIDS also significantly inhibits the growth rate of the whole bone’s length, although its inhibitory effect appeared to be lower than that of EIPA.

Although DIDS was previously considered asa non-specific inhibitor of AE2 because it was reported to be a potent chloride channels blocker, a recent review by Wulff, indicates that DIDS is a specific inhibitor of AE2 [44,45]. It was also reported by Liu et al. [25] and Ragel et al. [46] that DIDS specifically inhibit anion exchange by blocking extracellular chloride ion in exchange for HCO_3_^−^. Based on these facts, DIDS is now considered a specific AE2 inhibitor.

Moreover, an apparent inhibition of the total GP length was observed in the tibial bone when treated with EIPA or DIDS, but there were no significant differences with the control. This suggests that longitudinal bone growth is probably not determined by the total GP length but is limited to active physiological activities of some specific zone(s) of GP chondrocytes. It was widely reported by many authors that the hypertrophic chondrocytes zone is the major determinant of long bone growth [2,10,19,40].

This study also revealed significant differences in GP HCZ length in both EIPA and DIDS treatments with their respective controls. This implies that the hypertrophic chondrocytes zone could be the major determinant of long bone growth. This finding is in line with a previous study reported by Loqman et al. [10]. Our finding also showed that there were significant differences in the hypertrophic chondrocyte densities in both EIPA and DIDS treatment with their corresponding control groups. This finding, along with the significant HCZ length variation among the treated groups, may suggest that HCZ lengthening could be partly associated with the chondrocyte densities increase within the zone. The significant reduction in the hypertrophic chondrocytesnoticed in the plasma membrane inhibited treated groups might be associated with the cellular regulatory pH phenomenon that brings about the hypertrophic chondrocyte’s death, hence decreasing in chondrocyte density. Although the chondrocyte volume was not determined in this study, it was widely reported by many authors that hypertrophic chondrocyte volume increase is the major determinant of long bone growth [10,17,19,47].

Furthermore, it was observed that both NHE1 and AE2 membrane proteins were localized along the GP length in the control groups with an apparent decrease in fluorescence labeling associated with the membrane transporters in the treated groups (Figure 4b). The inhibited fluorescence labeling of NHE1 and AE2 was more prominent in the hypertrophic zone of the chondrocyte than in the proliferative chondrocytes zone. This may indicate that there are more physiological activities of the plasma membrane protein under investigation around the HCZ. On the other hand, the localization of the NHE1 and AE2 in the PCZ may suggest that they contribute to the other physiological functions during chondrocyte differentiation.

This study has demonstrated the contributions of NHE1 and AE2 in whole bone lengthening and growth plate regulation. Previous studies conducted by Bush et al. [19] have also demonstrated that the NKCC1 plasma membrane transporter equally has a role in whole bone lengthening and growth plate hypertrophic chondrocyte volume regulation that positively increases bone growth. In order to gain more insight into the role of NHE1 and AE2 membrane transport in longitudinal bone growth, there is a need for an extensive investigation using molecular techniques such as the use of small interfering (si) or short hairpin (sh) RNA-based knock down and other molecular procedures. Thus, this could be one of the limitations of the present study. Other limitations of this study that need to be pointed out include the inability to precisely determine the mineralized zone of the GP and the lack of labeling with apoptosis markers in order to determine the precise causes of the chondrocyte deaths, even though autophagy has also been recently implicated as the suspected cause of the hypertrophic chondrocyte deaths during longitudinal bone growth.

## 5. Conclusions

In summary, it can be concluded that Na^+^/H^+^ and HCO_3_^−^ exchange across the plasma membrane of growth plate chondrocytes could have an important role in longitudinal bone growth and growth plate lengthening (GP) regulation. The GP lengthening was majorly driven by an increase in the overall GP chondrocyte and hypertrophic chondrocyte densities. The possible mechanism behind the decreased chondrocyte densities observed in the groups that were treated with plasma membrane inhibitors could be associated to chondrocyte pH regulation following osmotic cell membrane exchange of the transporters under investigation. This may lead to hypertrophic chondrocyte deaths, hence a reduction in chondrocyte densities.

## Figures and Tables

**Figure 1 membranes-12-00707-f001:**
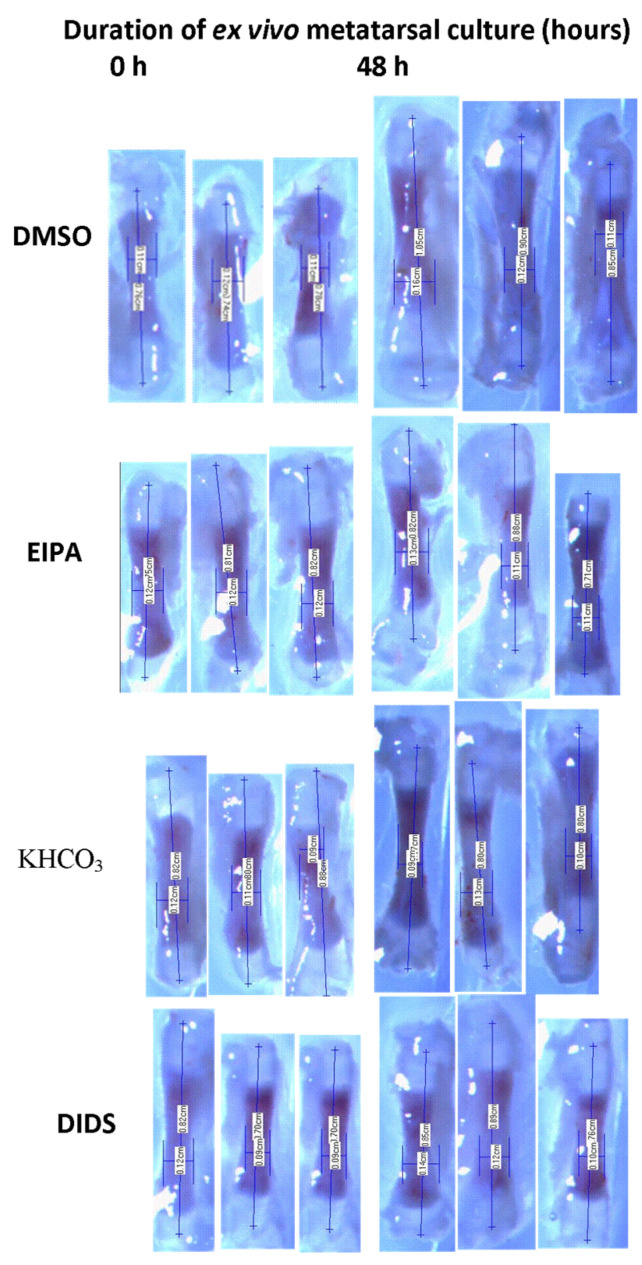
Representative of Serial stereomicroscopic images of metatarsal bone before and after treatments with EIPA and DIDS at respective concentrations of 444 µM and 250 µM. The lengths of the bones were measured in centimeter (cm) at X6.5 objective magnification.

**Figure 2 membranes-12-00707-f002:**
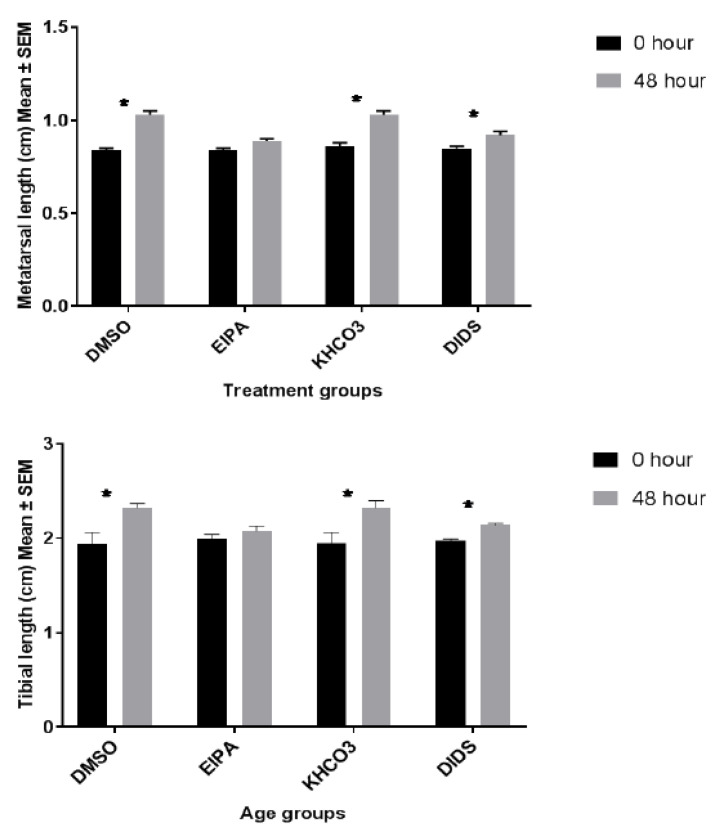
The bar chart shows metatarsal and tibial length (cm) growth changes at 0 and 48 h incubation with different treatments. Data were pooled from the left and right three middle metatarsals and their corresponding tibia from 20 rats. Each group represent fifteen metatarsal (*n* = 15) and ten tibial bones (*n* = 10). * indicates significant differences (*p* < 0.05; paired Student’s *t*-test) between baseline and the corresponding 48 h bone growth rates, data were expressed as means ± SEM.

**Figure 3 membranes-12-00707-f003:**
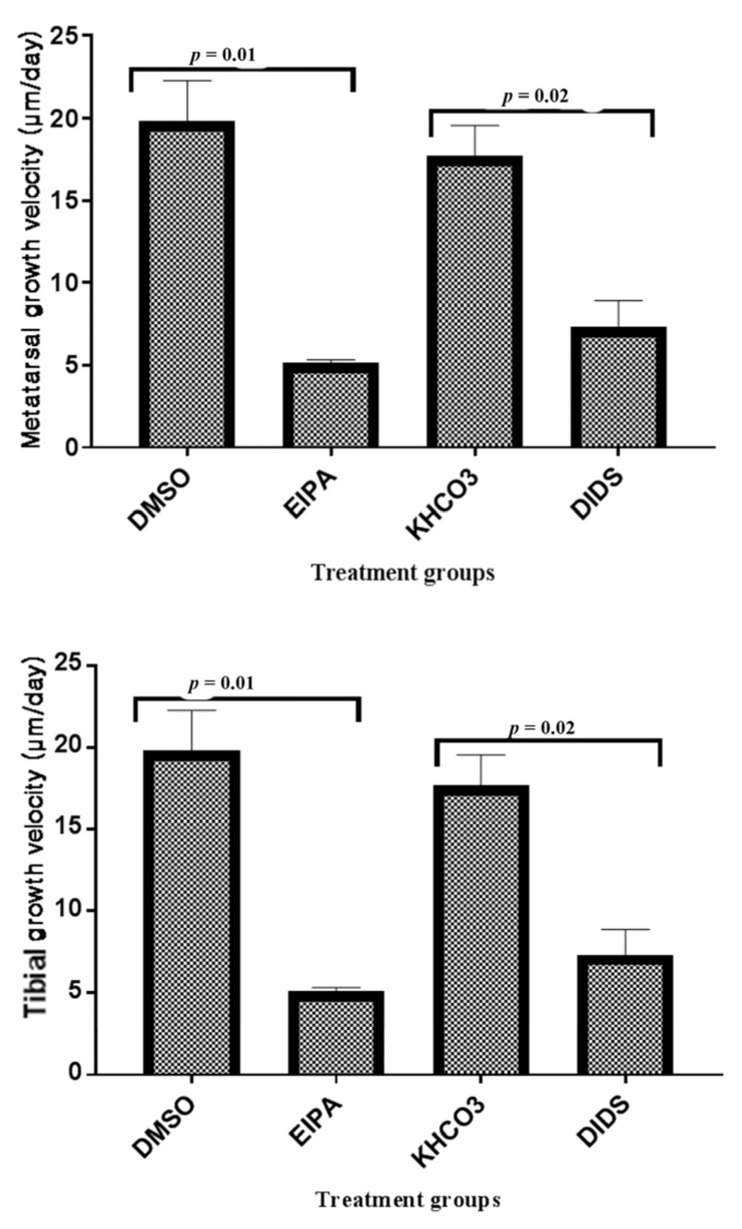
The bar chart shows the mean metatarsal and tibial growth velocity (μm/day) trend of changes between treated groups and their corresponding control. The *p*-value above the bars indicatesa significant difference between the treated and control groups (unpaired Student *t*-test).

**Figure 4 membranes-12-00707-f004:**
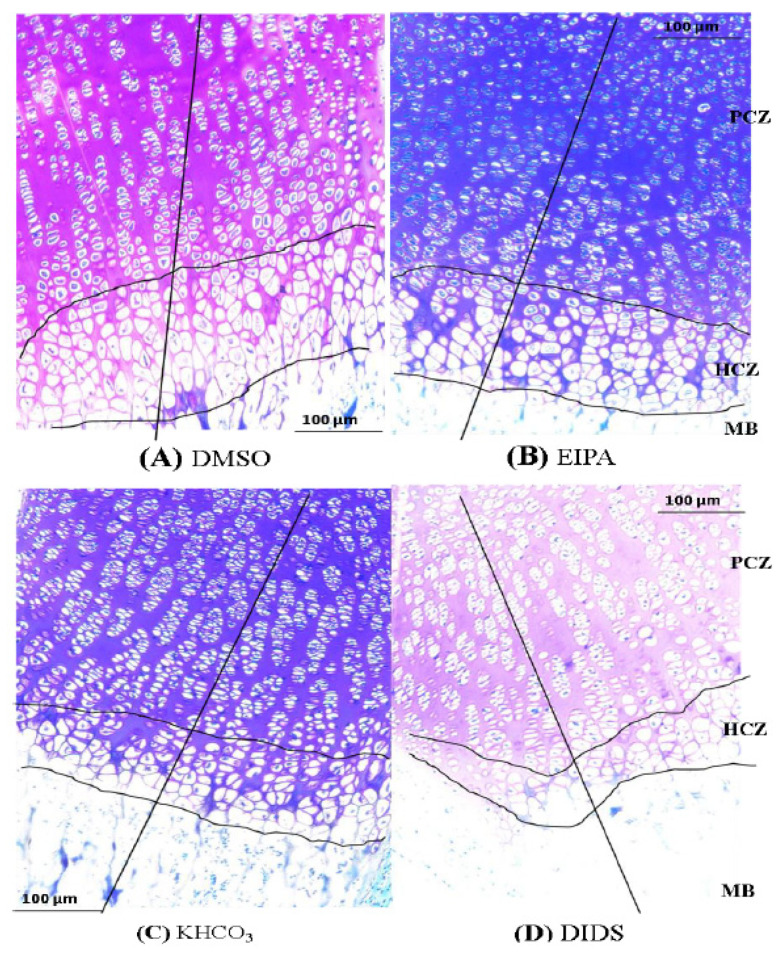
Histomicrograph showing the height of the tibial GP and HCZ of GP in different treatment groups after 48 h treatments with EIPA or DIDS. The straight vertical line at the -mid-GP section was used to determine the total GP length. Panel (**A**,**B**) are a comparison of HCZ length in EIPA-treated and its control group. Panels (**C**,**D**) are a comparison of HCZ length in DIDS-treated and its control group. Scale bar = 100 μm in all panels, X10 objective. PCZ, HCZ, and MB stand for proliferative chondrocytes zone, hypertrophic chondrocytes zone, and mineralized bone, respectively. Slides were stained with toluidine blue O.

**Figure 5 membranes-12-00707-f005:**
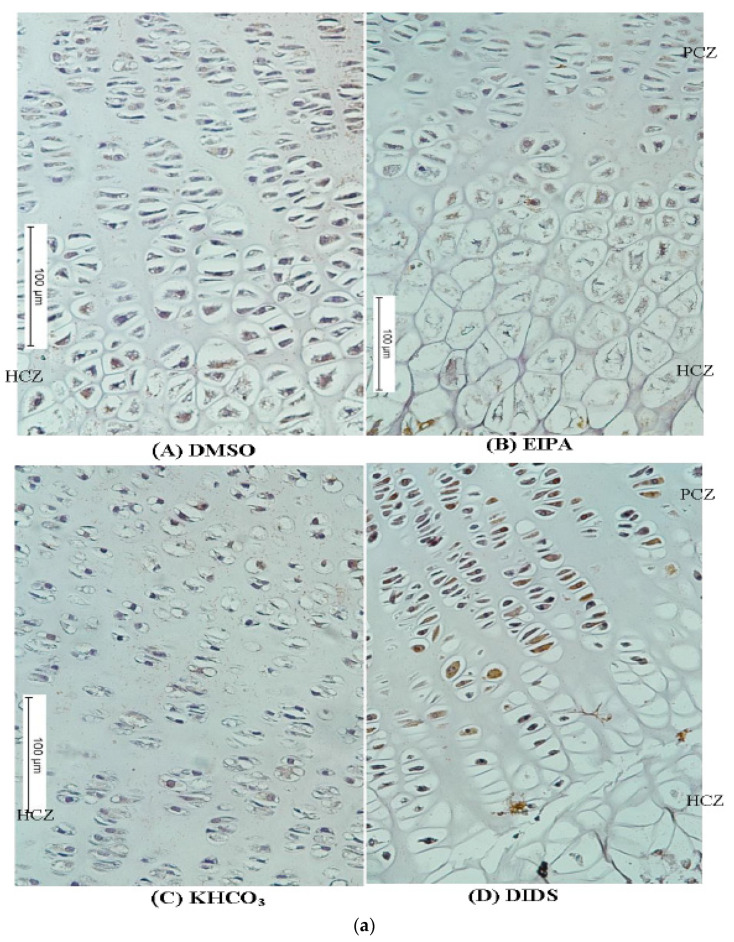
(**a**) Immunostaining micrographs showing the appearance of immunoperoxidase (IP) staining intensity of different treatment groups across the GP. Panels (**A**–**D**) are a comparison of immunoperoxidase localisations of N^+^/H^+^ or HCO_3_^−^ in the GP chondrocytes treated with EIPA or DIDS and their corresponding controls. Scale bar = 100 μm in all panels, X10 objective. (**b**) Immunofluorescence micrographs showing the appearance of fluorescence labeling intensity of different treatment groups across the GP. Panels (**A**–**D**) are the comparison of fluorescence labeling intensity in EIPA and DIDS with their respective control groups. Scale bar = 100 μm in all panels, X10 objective.

**Table 1 membranes-12-00707-t001:** Mean metatarsal and tibial percentage growth rates in different groups after 48 h ex vivo incubation.

Treatments	Metatarsal Growth Rate (%)	Tibial Growth Rate (%)
DMSO control	18.12 ±1.93 ^a^	15.63 ± 5.54 ^a^
EIPA (444 µM)	5.50 ± 0.96 ^b^	4.56 ± 0.755 ^b^
KHCO_3_ control	16.34 ± 1.49 ^b^	16.13 ± 3.85 ^b^
DIDS (250 µM)	7.43 ± 1.41 ^a^	7.52 ± 1.15 ^a^

Data were pooled from the right and left three middle metatarsals and their corresponding tibial bones from 20 rats. The percentage bone growth rates were expressed from bone lengthening after 48 h incubation. Treatments groups with different superscripts within the column significantly differ (*p* < 0.05; one-way ANOVA). Data were expressed as means ± SEM.

**Table 2 membranes-12-00707-t002:** The Effect of EIPA or DIDS treatments on total tibial GP length, HCZ length, and percentage HCZ length.

Treatments	Total GP Length (µm)	HCZ Length (µm)	HCZ (% of Total)
DMSO control (*n* = 5)	632 ± 33 ^b^	143 ± 5 ^ab^	24.33 ± 1.75 ^a^
EIPA (444 µM; *n* = 5)	516 ± 30 ^ab^	129 ± 4 ^b^	23.08 ± 1.32 ^a^
KHCO_3_ control (*n* = 5)	603 ± 33 ^ab^	137 ± 6 ^ab^	23.95 ± 0.94 ^a^
DIDS (250 µM; *n* = 5)	510 ± 29 ^a^	121 ± 5 ^a^	23.21 ± 1.50 ^a^

Different ^a,b^ Superscript within the columns indicates significant differences among treatment groups (*p* < 0.05; one-way ANOVA). Data were expressed as means ± SEM.

**Table 3 membranes-12-00707-t003:** The effect of EIPA or DIDS treatments on tibial total GP and HCZ cell densities.

Treatments	Total GP Chondrocyte Densities (Cells/mm^2^)	HCZ Chondrocyte Densities (Cells/mm^2^)
DMSO control (*n* = 5)	2409 ± 21 ^b^	1122 ± 16 ^b^
EIPA (444 µM; *n* = 5)	1609 ± 40 ^a^	742 ± 18 ^a^
KHCO_3_ control (*n* = 5)	2500 ± 22 ^a^	1151 ± 13 ^a^
DIDS (250 µM; *n* = 5)	1656 ± 33 ^c^	789 ± 10 ^b^

Different ^a,b,c^ Superscripts within the columns denote significant differences (*p* < 0.05; one-way ANOVA) among the treatment groups. Data were expressed as means ± SEM.

**Table 4 membranes-12-00707-t004:** Immunohistochemistry (IHC) score of different treatment groups after 48 h ex vivo incubation.

Treatments	Number of Slides Viewed (*n*)	Median IP Scores	*p* Values
DMSO; (*n* = 4)	12	4.0	0.001
EIPA (444 µM); (*n* = 4)	12	1.0	
KHCO_3_; (*n* = 4)	12	3.5	0.001
DIDS (250 µM); (*n* = 4)	12	1.0	

Data were generated from proximal GP of tibias from sixteen (16) rats, and 3 different fields of interest were viewed from each slide. The *p*-values indicate a significant difference (*p* < 0.05; non-parametric Mann–Whitney *t*-test) between the treatment and control groups.

## Data Availability

The data generated from this study are available from the corresponding author upon reasonable request provided it will be judiciously used.

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
