# Peer review of "Roles of Sodium Hydrogen Exchanger (NHE1) and Anion Exchanger (AE2) across Chondrocytes Plasma Membrane during Longitudinal Bone Growth"

_membranes, 2022, doi:10.3390/membranes12070707_

Round 1

Reviewer 1 Report

The clearly written review “Role of sodium hydrogen exchange (NHE1) and anion exchanger (AE2) across chondrocytes plasma membrane during longitudinal bone growth” deals with possible mechanisms involved in longitudinal bone growth. The presented data are solid and conclusive, but so far, only a summary of measurements. In order to make the manuscript more interesting for the scientific community it would be necessary to add more functional data, especially for a possible mechanism.

Furthermore:

It would be of interest to measure the volume of the chondrocytes dependent on the presence of the inhibitors.

What is the reason for the decreased labeling intensity of the transporters in the presence of the inhibitors (fig. 4)? What is the mechanism behind?

Fig. 4: It would be of importance to perform; in addition, a labelling with an apoptosis marker or to perform a tunnel assay.

There is also a discrepancy as it is known that DIDS is also an inhibitor for caspase activity. How would the authors explain this fact in context with their results?

Author Response

It would be of interest to measure the volume of the chondrocytes dependent on the presence of the inhibitors.

Answer: Previous study conducted by Loqman et al. 2013 (a principle investigator of this study) have shown that chondrocyte volume have significant role to play during ex vivo bone growth with dose dependent exposure of similar inhibitors (EIPA and DIDS). Therefore, this study pays more emphasis on the roles of chondrocytes density within the growth plate.

What is the reason for the decreased labeling intensity of the transporters in the presence of the inhibitors (fig. 4)? What is the mechanism behind?

Answer: The decrease in labeling intensity of the transporters in the presence of the inhibitors may signify decreased cellular physiological functions leading to either decrease in chondrocytes volume or decrease chondrocytes density, associated to cells death or both. This in turn lead to significant bone growth inhibition noticed in the treated groups. The likely mechanism behind the regulatory volume phenomenon or decreased chondrocytes density could be due to a reduction to chondrocyte volume set-point as result of chondrocyte pH regulation following osmotic cell membrane exchange of transporters under investigation. The cellular pH regulation could also lead to cell dead, depending on the concentration of the inhibitors, which may bring about decrease in chondrocytes density at a given point as noticed in this case.

Fig. 4: It would be of importance to perform; in addition, a labeling with an apoptosis marker or to perform a tunnel assay.

Answer: That is right, labeling with apoptosis marker or tunnel assay would have help in differentiating the specific cause of the chondrocytes death, but unfortunately this was not part of the initial study design. This could also be part of the limitations of this study and is going to be pointed out in the course of discussion. It is equally important to note that autophagy have also been implicated in hypertrophic chondrocytes death during longitudinal bone growth.  

There is also a discrepancy as it is known that DIDS is also an inhibitor for caspase activity. How would the authors explain this fact in context with their results?

Answer: That is right, some studies have shown that DIDS inhibit caspase activity which serve as a good biomarker of apoptosis, but this occur at high concentration of DIDS at about 500µM. It was also not proven that DIDS is a specific inhibitor of caspase as compared to z-VAD-fmk, qVD-oph and IDN-6556. It is also not clear and still debatable if apoptosis or autophagy is responsible for death of the hypertrophic chondrocytes during endochondral ossifications to create a space for advancing bone growth. At the concentration 250µM of DIDS we used, apoptosis may not be inhibited.

Reviewer 2 Report

This article, entitled "Roles of sodium hydrogen exchanger (NHE1) and anion exchanger (AE2) across chondrocytes plasma membrane during longitudinal bone growth" investigates the roles of sodium hydrogen exchanger [NHE1]) and anion exchanger [AE2]) during longitudinal bone growth in rats by culturing metatarsals from 10 days-old rats in the presence or absence of NHE1 and AE2 inhibitors. The authors hypothesized that Na+/H+ and HCO3 plasma membrane transports of chondrocytes have a role in mediating longitudinal bone growth through other cellular mechanisms apart from the hypertrophic chondrocytes cell volume regulation. The results obtained by the authors revealed that culture of the bones in the presence of inhibitors significantly reduced bone growth, bone growth velocity and the length of hypertrophic chondrocyte zone without significant effect on the total growth plate length.

The paper is interesting since the information provided add new information about the cellular processes underlying bone growth. The culture of metatarsals is a method that allows the analysis of numerous interesting parameters. I have only two minor points to improve the manuscript:

1. Measurements of the mineralized regions of the cultured metatarsals can be easily performed with standard phase contrast microscopy and image analysis software. Data on mineralization can complement those obtained on the growth plate.

2. A figure of serial images of treated and untreated metatarsals at low magnification can be very informative.

Author Response

  1. Measurements of the mineralized regions of the cultured metatarsals can be easily performed with standard phase contrast microscopy and image analysis software. Data on mineralization can complement those obtained on the growth plate.

Answer: An attempt was made to measure the mineralized zone from the histomorphometric with Image J, but it was not giving us accurate and reliable results; as such this was excluded from the initial design. This is going to be mention as one of the limitations of this study.

  1. A figure of serial images of treated and untreated metatarsals at low magnification can be very informative.

Answer: Stereo-microscopic images before and after treatment were added as figure 1, as suggested. The title of the figure read “Figure 1: Representative of Serial stereo-microscopic images of metatarsal bone before and after treatments with EIPA and DIDS at respective concentrations of 444 µM and 250 µM. The lengths of the bones were measured in centimeter (cm) at X6.5 objective magnification”.  

Round 2

Reviewer 1 Report

The manuscript is still very descriptive and speculative. The changes between the old and new version of the manuscript are minimal. There is still the need for more functional data in order to support the proposed mechanism of the authors. Where is the problem to perform further experiments like a tunnel assay or a labeling with a marker for apoptosis. In addition, the authors speculated about possible pH changes in the chondrocytes. Why not performing an assay detecting changes of pH. All these experiments would really strengthen the manuscript.

Author Response

Response to Reviewer

The manuscript is still very descriptive and speculative. The changes between the old and new version of the manuscript are minimal. There is still the need for more functional data in order to support the proposed mechanism of the authors. Where is the problem to perform further experiments like a tunnel assay or a labeling with a marker for apoptosis. In addition, the authors speculated about possible pH changes in the chondrocytes. Why not performing an assay detecting changes of pH. All these experiments would really strengthen the manuscript.

We have acknowledged and appreciate the quality of your comments and observations. There is no doubt that the tunnel assay/apoptosis marker recommended will surely improve the quality of our manuscript. However, we could not do it due to lack of funding, because we have exhausted the funds allocated for this research, as such we are not in a position to undertake any further research to address these issues. We have pointed out all these issues as limitations of the current study, we hope those issues raised may serve as research problems to numerous researchers to investigate on.